# MSBoost: Using Model Selection with Multiple Base Estimators for Gradient Boosting

## Abstract

Gradient boosting is a widely used machine learning algorithm for tabular regression, classification and ranking. Although, most of the open source implementations of gradient boosting such as XGBoost, LightGBM and others have used decision trees as the sole base estimator for gradient boosting. This paper, for the first time, takes an alternative path of not just relying on a static base estimator (usually decision tree), and rather trains a list of models in parallel on the residual errors of the previous layer and then selects the model with the least validation error as the base estimator for a particular layer. This paper has achieved *state-of-the-art* results when compared to other gradient boosting implementations on 50+ tabular regression and classification datasets. Furthermore, ablation studies show that MSBoost is particularly effective for small and noisy datasets. Thereby, it has a significant social impact especially in tabular machine learning problems in the domains where it is not feasible to obtain large high quality datasets.

## 1 Introduction

Gradient boosting (Friedman, 2001; 2002) has been a powerful boosting (Schapire, 1990) based machine learning algorithm that has achieved state-of-the-art accuracy in various real world tasks. Such as in particle physics, biochemistry, finance, fraud detection, search engine recommendations, drug discovery and many others (Chen & He, 2015; Wu et al., 2018; Nobre & Neves, 2019; Hajek et al., 2023; Burges, 2010; Li et al., 2007; Gulin et al., 2011; Sikander et al., 2022; Sun et al., 2020; Natekin & Knoll, 2013; Roe et al., 2005; Wu et al., 2010; Zhang & Haghani, 2015). Its significance lies in its ability to handle diverse data types and complex feature engineering whilst effectively managing high-dimensional, noisy datasets with heterogeneous features.

It builds a 'stronger' predictive model by combining several weaker models through an iterative greedy process that focuses on correcting the errors of previous models, which is based on sound theoretical evidence as per (Kearns & Valiant, 1994). Popular implementations of gradient boosting include XGBoost (Chen & Guestrin, 2016), which enhances traditional methods by introducing regularization to prevent overfitting and tree pruning to improve efficiency, and LightGBM (Ke et al., 2017), which differs by using a leaf-wise tree growth strategy instead of level-wise growth, and implements Gradient-based One-Side Sampling (GOSS) to speed up training on large datasets while maintaining accuracy. Furthermore, other variants include CatBoost (Prokhorenkova et al., 2018) which introduces a novel categorical encoding method to mitigate target leakage, and using Artificial Neural Network, Principal Component Analysis and Random Projections for feature extraction and combine this with gradient boosting as per AugBoost (Tannor & Rokach, 2019).

The main contribution of this paper, Model Selection based Gradient Boosting (MSBoost[1]), is to explore, for the first time, the usage of model selection in order to find the base estimator with the least validation error. Unlike the current methods which use a single base estimator, usually decision tree (Li et al., 1984; Friedman et al., 2000; Rokach & Maimon, 2005), although previous research has been done in boosting other models (Zięba et al., 2014). Benchmarking this method, MSBoost, on 50+ datasets indicate that this method outperforms previous methods such as LightGBM and XGBoost, and based on the ablation studies performed

---

[1] `https://github.com/AnnonAIResearcher/MSBoost/tree/main/MSBoost-main`

it can be observed that MSBoost is particularly effective for small and noisy datasets. Thereby, MSBoost would be particularly effective for tabular regression and classification problems where it is not feasible or expensive to obtain thousands of high quality samples.

## 2 Method

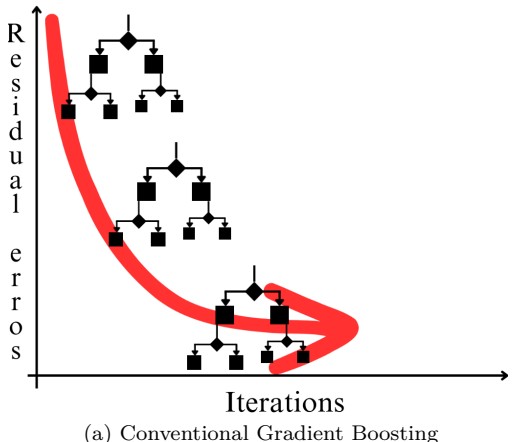

(a) Conventional Gradient Boosting

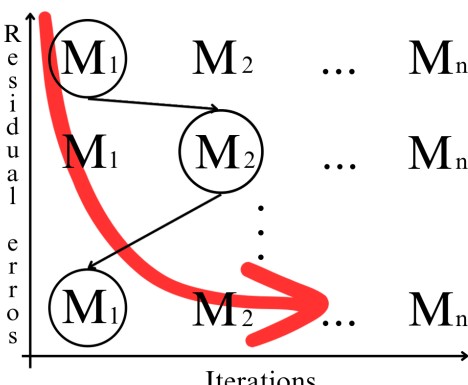

(b) Model Selection based Gradient Boosting (MS-Boost)

Figure 1: Conventional Gradient Boosting methods usually use Decision Trees, also known as CART(s), as the sole base estimator in order to minimize the residual errors over a number of iterations. Whereas, MSBoost from a list of ML models dynamically would choose the one with the least residual errors, in parallel, and use it as the base estimator for that layer.

Similar to gradient boosting, the goal of MSBoost is to approximate any arbitrary but particular $\mathcal{F} : \mathbb{R}^m \to \mathbb{R}$ with a series of additive and scaled $F_i$ in order to minimize $\mathcal{L}(\mathcal{F}(\mathbf{x}), F(\mathbf{x}))$. For any given tabular dataset $\mathcal{D} = \{(\mathbf{x}_i, y_i)\}_{i=1}^n$, and a differentiable loss function $\mathcal{L}(\mathbf{y}, F(\mathbf{x}))$. Wherein $\mathbf{x}_i$ is an arbitrary but particular vector $\mathbf{x}_i = (x_i^1, x_i^2, \ldots, x_i^m)$ containing $m$ features, and $\mathbf{y} \in \mathbb{R}^n$, which has $n$ samples is the target vector. First, MSBoost initializes the first estimator as a constant term i.e $F_0(\mathbf{x}) = \arg\min_k \sum_{i=1}^n \mathcal{L}(y_i, k)$, which turns out to be the arithmetic mean of the target values vector $\mathbf{y}$. Next, for each subsequent iteration $i = 1, \ldots, N$ it shall compute the pseudo residuals:

$$\mathbf{r}_i = -\left[\frac{\partial \mathcal{L}(\mathbf{y}, F_{i-1}(\mathbf{x}))}{\partial F_{i-1}(\mathbf{x})}\right] \tag{1}$$

and the base estimator for $i^{\text{th}}$ layer is based on a list of models $\mathcal{M}$, such that:

$$h_i(\mathbf{x}) = \arg\min_{\forall M \in \mathcal{M}} \mathcal{L}(\mathbf{y}, M(\mathbf{r}_i))(\mathbf{r}_i) \tag{2}$$

Finally it would update the model for $i^{th}$ layer, i.e $F_i(\mathbf{x}) = F_{i-1}(\mathbf{x}) + \alpha \cdot h_i(\mathbf{x})$, and the final prediction, $\hat{\mathbf{y}} = F(\mathbf{x}) = F_0(\mathbf{x}) + \sum_{i=1}^N F_i(\mathbf{x})$.

---

**Algorithm 1** MSBoost Algorithm Pseudocode

---

**Require:** Tabular dataset $\mathcal{D} = \{(\mathbf{x}_i, y_i)\}_{i=1}^n$, differentiable loss function $\mathcal{L}(\mathbf{y}, F(\mathbf{x}))$, models $\mathcal{M}$, number of iterations $N$, learning rate $\alpha$

1: Initialize $F_0(\mathbf{x}) = \arg\min_k \sum_{i=1}^n \mathcal{L}(y_i, k)$
2: **for** $i = 1$ to $N$ **do**
3:    $\mathbf{r}_i = - \left[ \frac{\partial \mathcal{L}(\mathbf{y}, F_{i-1}(\mathbf{x}))}{\partial F_{i-1}(\mathbf{x})} \right]$
4:    $h_i(\mathbf{x}) = \arg\min_{\forall M \in \mathcal{M}} \mathcal{L}(\mathbf{y}, M(\mathbf{r}_i))(\mathbf{r}_i)$
5:    $F_i(\mathbf{x}) = F_{i-1}(\mathbf{x}) + \alpha \cdot h_i(\mathbf{x})$
6: **end for**
7: **return** $\hat{\mathbf{y}} = F_0(\mathbf{x}) + \sum_{i=1}^N F_i(\mathbf{x}) = 0$

---

## 2.1 Rationale for Model Selection in Gradient Boosting

Since model selection searches for $\arg\min_{M \in \mathcal{M}} \mathcal{L}(\mathbf{y}, M(\mathbf{r}_i))$ for each iteration $i$, $\Rightarrow \mathcal{L}(\mathbf{y}, M(\mathbf{r}_i)) \leq \mathcal{L}(\mathbf{y}, S(\mathbf{r}_i)), \forall S$ which are static machine learning models say Decision Tree. And over a large number of iterations $N$, $\mathbf{r}_{i,\mathcal{M}}$ (model selection, a dynamic method) $< \mathbf{r}_{i,S}$ (for any static base estimator). This is technically a "$\leq$" inequality, but based on the inductive proposition that over a large number of iterations, $N$, a static method would have higher $\mathbb{E}(\mathbf{r}_i)$ than dynamically selecting base estimators in each iteration, the "$<$" inequality should hold true. Wherein the base case is $\mathbb{E}(\mathbf{r}_{i,\mathcal{M}}) < \mathbb{E}(\mathbf{r}_{i,S})$, which is empirically true as per (Caruana & Niculescu-Mizil, 2006; McElfresh et al., 2024) and theoretically justified by the *No Free Lunch Theorem* (Wolpert & Macready, 1997; Wolpert, 2002). Furthermore, analysing the effect of specific base estimators stacked over $N$ iteration on the residual plots shall be an interesting obsevation, for example a non-linear model like (Cortes & Vapnik, 1995) may have a more linear residual plot when compared to that of a linear model, so a non linear base estimator in $i^{th}$ iteration may lead to a linear model in $i + 1^{th}$ iteration. But this has been left for a avenue for future research.

Also, as empirically demonstrated by (Caruana & Niculescu-Mizil, 2006; McElfresh et al., 2024), there is no *one-size-fits-all* baseline model which does well on all types of datasets, which empirically justifies as to why boosting multiple estimators might be effective; and, increase the diversity of the base learners, which potentially help to improve the generalization performance (i.e less variance) (Zhou, 2012).

## 2.2 Model Selection Methods[2]

**Naïve Method**  The naïve way for model selection is to train all the available base estimators on $\mathbf{r}_i$ in parallel. This way would ensure that the model with the least residual errors is truly being selected for each layer and precisely conforms to the theoretical rationale stated in Section 2.1. But this would have the largest time complexity, i.e $O$(Number of Iterations $\times$ Base model with the highest time complexity i.e the limiting factor).

**Random Sampling**  Sampling a subset of models from $\mathcal{M}$, shall reduce the overall training time, but this may not find the model with least possible validation residual errors.

**Frequency & Probability Based Sampling**  Assuming that only a subset of models from $\mathcal{M}$ would be used for most of the time due to the characteristics of the dataset being used. For the first $I$ iterations, this shall be a track of the frequency of the top $N$ models, and for the rest of the iterations only train the top $N$ models initially found. Here $I$ and $N$ are hyper-parameters. A more vigorous method for this would be to use Bayesian model selection (Wasserman, 2000; Ando, 2011; Chipman et al., 2001) using Dirchlet's distribution (Dirichlet, 1889; Dirichlet & Seidel, 1900) and train the models with the top $N$ probabilities of being used. (See Algorithm 2)

---

[2]The model selection was done on a validation dataset, subsampled from the training data.

---

**Algorithm 2** Update Posterior Probabilities for Models

---

**Require:** New observed error values $\mathbf{E}$, Prior probabilities for all models $\mathbf{P}$ (For the first iteration it is assumed that all models have an equal prior probability.), Indices of trained models $T$, Dirichlet prior parameters $\alpha$, Penalty factor $\beta = 0.7$

1: $\mathbf{P_T} \leftarrow [P_i \mid i \in \mathbf{T}]$
2: $\mathbf{S} \sim \text{Dir}(\boldsymbol{\alpha})^{1000}$
3: $\mathbf{W} \leftarrow []$      // Initialize weights
4: **for** $\mathbf{s} \in \mathbf{S}$ **do**
5:      $w \leftarrow \exp\left(-\sum_{i=1}^{n} \log(s_i) \cdot E_i\right)$      // Get probabilities
6:      $\mathbf{W} \leftarrow \mathbf{W} \cup \{w\}$
7: **end for**
8: $\mathbf{W} \leftarrow \frac{\mathbf{W}}{\sum \mathbf{W}}$
9: $\mathbf{P'_T} \leftarrow \mathbf{P_T} \cdot (\mathbf{W} \cdot \mathbf{S})$
10: $\mathbf{P'} \leftarrow \mathbf{P}$      // Initialize updated posterior probabilities
11: **for** $i \in \mathbf{T}$ **do**
12:      $P'_i \leftarrow \mathbf{P'_T}[i]$      // Update posterior probabilities for trained models
13: **end for**
14: $\mathbf{U} \leftarrow \{i \mid i \notin \mathbf{T}\}$
15: **for** $i \in \mathbf{U}$ **do**
16:      $P'_i \leftarrow P'_i \cdot \beta$      // Penalize untrained models
17: **end for**
18: **return** $\mathbf{P'} \leftarrow \frac{\mathbf{P'}}{\sum \mathbf{P'}}$ =0

---

## 3 Experiments & Discussion

**Comparison with baselines**   MSBoost (random sampling half of the models from $\mathcal{M}$ for training in each iteration) was compared[3] with XGBoost and LightGBM. The source code of the experiments are available, and can be reproduced (`https://github.com/AnnonAIResearcher/MSBoost/tree/main/MSBoost-main`). Unfortunately due to constrained computational resources the benchmarking was done on 1K samples on OpenML Vanschoren et al. (2014); Bischl et al. (2021) datasets with 0.01 lasso threshold to screen for irrelevant features which would have increased the computational costs. Table 1 compares the mean squared error with 5 fold cross validation (CV), and Table 2 compares the log loss with 5 Fold CV; please check Appendix A.1 for entire results. Paired single tailed $t$-test reveal that MSBoost yields a statistically significant improvement over LightGBM and XGBoost in metrics, with $p$-value $<< 0.001$ (excluding outliers like wave_energy), and $p < 0.02$ for standard deviation thus improving the bias-variance trade-off Briscoe & Feldman (2011). It should be noted even without regularization, and GOSS and EFB of XGBoost and LightGBM respectively, MSBoost has a statistically significant improvement. Thereby, this may have even better improvement over previous methods if those techniques are incorporated in MSBoost. (The complete results can be found in Appendix A.1, and Appendix B for source and description of these datasets.)

**Impact of dataset dependent factors**   Figure 2 highlights how MSBoost and the baseline models perform when noise, number of samples and others are progressively increased on Scikit-Learn's Pedregosa et al. (2011) make classification dataset Guyon (2003). This is a *cherry-picked* example, but similar trend was found on all other Scikit-Learn's synthetic datasets, their plots can be found in Appendix A.2. Using paired single tail $t$-test that MSBoost has a $p$-value $< 0.01$ when compared to XGBoost and LightGBM for robustness against noise and for impact of number of samples when compared to the baseline models.

---

[3]For now it wasn't compared to CatBoost, since in order to have a fair comparison, since MSBoost's implementation doesn't have targeted feature encoding for now.

Table 1: Comparison (regression) with baselines based on mean squared error (MSE)

|  | MSBoost | LightGBM | XGBoost |
|---|---|---|---|
| wave_energy | 0.0 ± 0.0 | 1.9e+9 ± 2.9e+8 | 3.0e+9 ± 4.5e+8 |
| Friedman 2 | 150 ± 31 | 385 ± 57 | 501 ± 58 |
| Sparse Uncorr. | 1.0 ± 0.15 | 1.5 ± 0.11 | 1.7 ± 0.22 |
| kin8nm | 2.1e-2 ± 1e-4 | 3.1e-2 ± 1.5e-3 | 3.6e-2 ± 1.3e-3 |
| sarcos | 32 ± 8 | 46 ± 15 | 48 ± 10 |
| Moneyball | 431 ± 24 | 588 ± 42 | 635 ± 39 |
| yprop_4_1 | 7e-4 ± 1e-4 | 9e-4 ± 1e-4 | 1.1e-3 ± 1e-4 |
| fps_benchmark | 2354 ± 110 | 2917 ± 104 | 3758 ± 395 |
| Zurich Transport | 10 ± 0.7 | 12 ± 0.9 | 15 ± 1.4 |
| Diabetes | 3017 ± 333 | 3590 ± 433 | 3991 ± 651 |

Table 2: Comparison (classification) with baselines based on log loss

|  | MSBoost | LightGBM | XGBoost |
|---|---|---|---|
| phoneme | 0.34 ± 0.03 | 0.43 ± 0.07 | 0.43 ± 0.06 |
| guillermo | 0.56 ± 0.04 | 0.69 ± 0.10 | 0.77 ± 0.11 |
| MagicTelescope | 0.40 ± 0.04 | 0.48 ± 0.05 | 0.50 ± 0.05 |
| heloc | 0.58 ± 0.01 | 0.67 ± 0.08 | 0.78 ± 0.09 |
| Bioresponse | 0.50 ± 0.02 | 0.57 ± 0.08 | 0.59 ± 0.07 |
| electricity | 0.54 ± 0.06 | 0.61 ± 0.08 | 0.65 ± 0.09 |
| Australian | 0.50 ± 0.03 | 0.54 ± 0.06 | 0.64 ± 0.08 |
| house_16H | 0.38 ± 0.03 | 0.40 ± 0.07 | 0.42 ± 0.06 |
| pol | 0.17 ± 0.04 | 0.18 ± 0.05 | 0.15 ± 0.04 |
| california | 0.37 ± 0.03 | 0.39 ± 0.05 | 0.40 ± 0.06 |

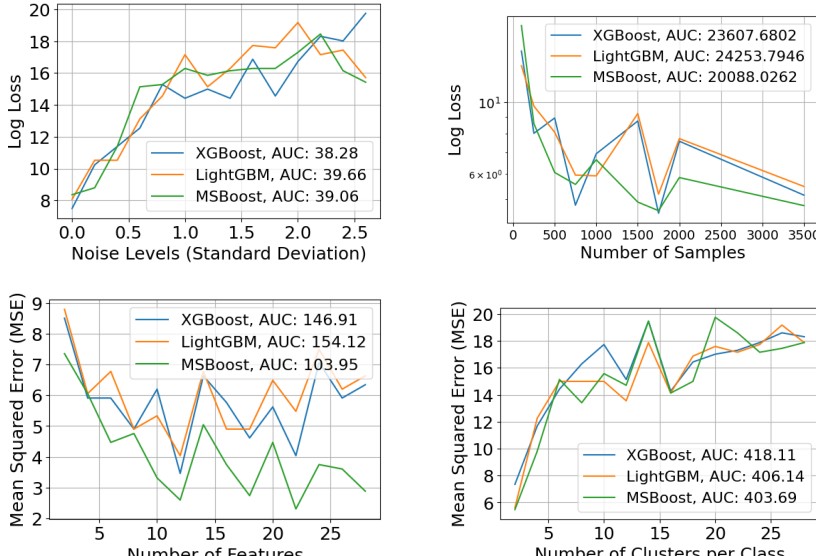

Figure 2: Impact of dataset dependent various factors on log loss for Make Classification Dataset Guyon (2003)

**Impact of model selection methods**  The effect of number of base models trained on the model selection methods is demonstrated in Figure 3, this is a *cherry-picked* example the rest can be found in Appendix A.3. There is no statistically significant difference in choosing the bayes method over the frequency based method ($p = 0.28$), but the bayes method turns out to be better than random sampling ($p = 0.06$).

Table 4: $p$-values for impact of number of models on model selection methods (Row vs. Column)

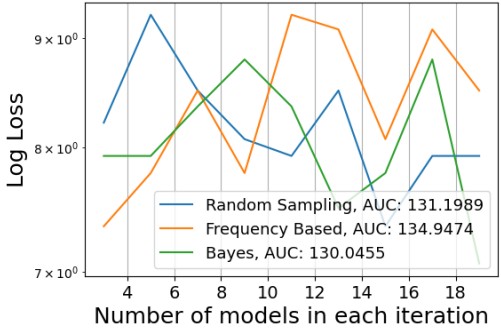

Figure 3: Impact of changing number of models trained, for model selection methods (Scikit-Learn's Make Classification dataset)

Table 3: Comparison on small noisy real world datasets with significant social impact ([†]MSE & [‡]Log Loss)

| | UCI ID Kelly et al. | MSBoost | LightGBM | XGBoost |
|---|---|---|---|---|
| AIDS Clinical Trials[†] | 890 | 8.1e-2 ± 6.5e-3 | +7.1% ± +33.8% | +13.5% ± -1.5% |
| Student Performance[†] | 320 | 6.3 ± 0.81 | +5.4% ± +15.3% | +25.9% ± +42.2% |
| Energy Efficiency[†] | 242 | 1.7 ± 0.18 | +2.2% ± +16.9% | +34.3% ± +19.1% |
| Diabetes[†] | Pedregosa et al. (2011) | 3017 ± 333 | +18.9% ± +30.0% | +32.9% ± +95.4% |
| Liver Disorders[†] | 60 | 9.6 ± 1.19 | +4.7% ± +2.6% | +23.1% ± -0.3% |
| Heart Failure Clinical Records[†] | 519 | 0.12 ± 0.03 | +3.4% ± +35.95% | +17.9% ± +37.1% |
| Thyroid Cancer Recurrence[‡] | 915 | 1.2 ± 0.88 | +30.6% ± -13.1% | +22.9% ± +4.1% |
| Rice (Cammeo and Osmancik)[‡] | 545 | 2.79 ± 0.10 | +7.7% ± +130.3% | +6.1% ± +31.0% |
| Blood Transfusion Service[‡] | 176 | 8.3 ± 0.48 | +10.4% ± +0.3% | +13.9% ± +148.0% |
| Acute Inflammations[‡] | 184 | 0.0 ± 0.0 | 0.3 ± 0.6 | 0.75 ± 1.2 |
| SPECTF Heart[‡] | 96 | 6.4 ± 1.24 | +4.2% ± +93.0% | +2.1% ± +34.9% |
| Glioma Grading Clinical & ...[‡] | 759 | 4.8 ± 0.75 | +33.9% ± +20.5% | +34.8% ± -26.7% |

| | Bayes | Frequency Based | Random Sampling |
|---|---|---|---|
| Bayes | 1.00 | 0.71 | 0.82 |
| Frequency Based | **0.28** | 1.00 | 0.82 |
| Random Sampling | **0.06** | 0.17 | 1.00 |

**Social Impact**  As mentioned above, there is statistically significant evidence that using model selection along with gradient boosting, MSBoost, may improve bias-variance trade-off. Particularly on small and noisy datasets, where usually other machine learning algorithms tend to overfit Lever et al. (2016); Oates & Jensen (1997). Table 3 demonstrates a few possible tabular regression and classification problems with significant social impact, where MSBoost turns out to be better than other methods in terms of MSE/log loss and standard deviation (5 Fold CV).

**Limitations & Further Prospects**  (i) Since it trains multiple models for each iteration, MSBoost, has a enormously high time complexity. Where the limiting factor is SVM's RBF kernel, which is quadratic. So the worst case time complexity of MSBoost is approximately $O(n^2)$, whereas LightGBM and others have a time complexity of $O(n \ log \ n)$ (ii) In theory MSBoost can be extended to multiclass classification using one-versus-rest or natively as well but for now MSBoost was not tested for multi-class classification. (iii) Due to system resource constrains (AMD Ryzen 5 3550H & 8 GB RAM, Ubuntu 20.4 & 22.04.4 LTS), and the enormous time complexity the test most of the benchmarking couldn't be done for more than 1K samples, although this was compensated by benchmarking on 50+ datasets with 5 fold CV. (iv) MSBoost, can be combined with regularization methods, adaptive learning rate, GOSS, EFB, and targeted feature encoding from XGBoost, LightGBM, and CatBoost, but for now this has been left for future work.

## 4  Conclusion

This paper introduces a novel gradient boosting method, MSBoost, which uses model selection to find base estimators for each iteration of gradient boosting. Empirical results show that there is a statistically significant evidence that this method outperforms other popular gradient boosting methods (LightGBM & XGBoost), both in terms of errors and standard deviation of the error. Furthermore, ablation studies reveal that MSBoost outperforms other methods on (synthetic & real) small and noisy datasets, a domain where machine learning algorithms usually struggle. Future work, shall incorporate techiques like targeted feature encoding, GOSS, EFB and other from the current Gradient Boosting methods.

### Broader Impact Statement

Our work significantly enhances the accuracy of classifiers and regressors, offering wide-ranging benefits across various fields. In robotics, improved predictive models can optimize motion control and decision-making, enabling more efficient automation. In healthcare, more accurate predictions can lead to earlier diagnoses, personalized treatment plans, and better patient outcomes. In finance, precise models can enhance risk assessment, fraud detection, and portfolio optimization, driving smarter investments and economic growth. However, this increased accuracy also presents risks, particularly the potential for malicious use. For instance, highly accurate models could be misused for unauthorized surveillance, discriminatory profiling, or financial market manipulation. In healthcare, they may overfit sensitive data, leading to privacy concerns or biased treatments. We underscore the need for ethical safeguards to mitigate these risks, advocating for fairness, transparency, and responsible use, in alignment with the TMLR Ethics Guidelines.

### Acknowledgments

Redacted for double-blind review

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

# A Extended Results

## A.1 Benchmarking Results

Table 5: Comparison (regression) with baselines based on mean squared error (MSE)

|  | MSBoost | LightGBM | XGBoost |
|---|---|---|---|
| wave_energy | 0.0 ± 0.0 | 1.979e+9 ± 2.967e+8 | 3.007e+9 ± 4.515e+8 |
| SGEMM_GPU_kernel_performance | 0.0006 ± 0.0001 | 0.0021 ± 0.0005 | 0.0005 ± 0.0001 |
| Friedman 2 | 150.5209 ± 31.6582 | 385.8898 ± 57.2398 | 501.6854 ± 58.3089 |
| Sparse Uncorrelated | 1.0338 ± 0.1538 | 1.5732 ± 0.1136 | 1.7742 ± 0.2214 |
| kin8nm | 0.0217 ± 0.0016 | 0.0318 ± 0.0015 | 0.0364 ± 0.0013 |
| sarcos | 32 ± 8 | 46 ± 15 | 48 ± 10 |
| Moneyball | 431.9247 ± 24.2184 | 588.8988 ± 42.4500 | 635.3445 ± 39.3275 |
| Parkinsons Telemonitoring | 13.5233 ± 2.1851 | 18.2569 ± 3.4753 | 13.1871 ± 1.8204 |
| yprop_4_1 | 0.0007 ± 0.0001 | 0.0009 ± 0.0001 | 0.0011 ± 0.0001 |
| fps_benchmark | 2354.4510 ± 110.9576 | 2917.2280 ± 104.0124 | 3758.8541 ± 395.4950 |
| Zurich Transport | 10.0047 ± 0.7051 | 12.3082 ± 0.9710 | 15.2101 ± 1.4006 |
| Diabetes | 3017.3830 ± 333.9345 | 3590.3865 ± 433.2183 | 3991.1318 ± 651.7501 |
| medical_charges | 0.0057 ± 0.0021 | 0.0067 ± 0.0019 | 0.0068 ± 0.0021 |
| Airlines_DepDelay_1M | 3.7258 ± 0.2437 | 4.3522 ± 0.3432 | 5.2531 ± 0.3028 |
| visualizing_soil | 24.9784 ± 6.9898 | 28.2629 ± 7.3992 | 22.4963 ± 9.5333 |
| video_transcoding | 146.3238 ± 51.3910 | 163.5462 ± 60.2620 | 206.6265 ± 64.0276 |
| health_insurance | 310.6828 ± 27.6937 | 345.2994 ± 31.4665 | 397.2537 ± 25.3154 |
| grid_stability | 0.0009 ± 0.0001 | 0.0010 ± 0.0001 | 0.0012 ± 0.0001 |
| abalone | 5.3366 ± 0.8634 | 5.8872 ± 1.2488 | 6.1480 ± 1.1375 |
| Liver Disorders | 9.1786 ± 1.1586 | 10.0795 ± 1.2242 | 11.8547 ± 1.1893 |
| student_performance_por | 8.1418 ± 1.3556 | 8.9132 ± 1.2743 | 12.5153 ± 1.6150 |
| diamonds | 1.93e+6 ± 3.44e+5 | 2.09e+6 ± 4.88e+5 | 2.30e+6 ± 4.59e+5 |
| auction_verification | 9.34e+7 ± 1.15e+7 | 1.00e+8 ± 9.79e+6 | 1.52e+8 ± 2.04e+7 |
| cpu_act | 10.2950 ± 1.5068 | 10.7118 ± 1.9016 | 16.6299 ± 10.0559 |
| Student Performance | 6.3849 ± 0.9935 | 6.5426 ± 0.9590 | 7.3635 ± 1.2074 |
| pol | 103.6665 ± 21.9906 | 105.9581 ± 42.7356 | 126.1512 ± 46.4249 |
| AIDS Clinical Trials Group Study | 0.0857 ± 0.0065 | 0.0868 ± 0.0087 | 13.1871 ± 1.8204 |
| Bike_Sharing_Demand | 12307.1564 ± 1427.9618 | 12461.6368 ± 1476.1868 | 13478.9580 ± 1027.8582 |
| srsd-feynman_hard | 2.549e-70 | 2.578e-70 | 2.984e-70 |
| seattlecrime6 | 151041.8081 ± 3619.6563 | 151809.1711 ± 3291.9561 | 151684.2628 ± 3180.8946 |

Table 6: Comparison (classification) with baselines based on log loss

|  | MSBoost | LightGBM | XGBoost |
|---|---|---|---|
| phoneme | 0.3467 ± 0.0371 | 0.4324 ± 0.0750 | 0.4393 ± 0.0623 |
| guillermo | 0.5644 ± 0.0461 | 0.6988 ± 0.1070 | 0.7725 ± 0.1146 |
| MagicTelescope | 0.4020 ± 0.0428 | 0.4817 ± 0.0510 | 0.5090 ± 0.0512 |
| heloc | 0.5888 ± 0.0130 | 0.6773 ± 0.0831 | 0.7849 ± 0.0932 |
| Bioresponse | 0.5012 ± 0.0264 | 0.5705 ± 0.0811 | 0.5921 ± 0.0752 |
| electricity | 0.5462 ± 0.0645 | 0.6137 ± 0.0818 | 0.6530 ± 0.0997 |
| Australian | 0.5087 ± 0.0318 | 0.5459 ± 0.0691 | 0.6432 ± 0.0853 |
| house_16H | 0.3847 ± 0.0361 | 0.4086 ± 0.0702 | 0.4254 ± 0.0681 |
| pol | 0.1738 ± 0.0489 | 0.1839 ± 0.0598 | 0.1524 ± 0.0468 |
| Bioresponse | 0.5431 ± 0.0805 | 0.5705 ± 0.0811 | 0.5921 ± 0.0752 |
| california | 0.3736 ± 0.0345 | 0.3911 ± 0.0552 | 0.4050 ± 0.0625 |
| heloc | 0.6507 ± 0.0872 | 0.6773 ± 0.0831 | 0.7849 ± 0.0932 |
| higgs | 0.7332 ± 0.1404 | 0.7543 ± 0.0862 | 0.8221 ± 0.1008 |
| compas-two-years | 0.6986 ± 0.1059 | 0.7138 ± 0.0543 | 0.8056 ± 0.0742 |
| Higgs | 0.7255 ± 0.0842 | 0.7409 ± 0.0540 | 0.8661 ± 0.0510 |
| MiniBooNE | 0.2995 ± 0.0294 | 0.3043 ± 0.0568 | 0.3087 ± 0.0610 |

Table 7: (Absolute values) Comparison on small noisy real world datasets with significant social impact ([†]MSE & [‡]Log Loss)

|  | **UCI ID** Kelly et al. | **MSBoost** | **LightGBM** | **XGBoost** |
|---|---|---|---|---|
| AIDS Clinical Trials[†] | 890 | 8.1e-2 ± 6.5e-3 | 8.7e-2 ± 1.1e-2 | 9.2e-2 ± 8.6e-3 |
| Student Performance[†] | 320 | 6.3 ± 0.81 | 6.6 ± 0.93 | 7.9 ± 1.15 |
| Energy Efficiency[†] | 242 | 1.7 ± 0.18 | 1.74 ± 0.21 | 2.28 ± 0.26 |
| Diabetes[†] | Pedregosa et al. (2011) | 3017 ± 333 | 3585 ± 433 | 4010 ± 651 |
| Liver Disorders[†] | 60 | 9.6 ± 1.19 | 10.05 ± 1.22 | 11.82 ± 1.18 |
| Heart Failure Clinical Records[†] | 519 | 0.12 ± 0.03 | 0.124 ± 0.041 | 0.141 ± 0.047 |
| Thyroid Cancer Recurrence[‡] | 915 | 1.2 ± 0.88 | 1.57 ± 0.77 | 1.47 ± 0.92 |
| Rice (Cammeo and Osmancik)[‡] | 545 | 2.79 ± 0.10 | 3.00 ± 0.23 | 2.96 ± 0.13 |
| Blood Transfusion Service[‡] | 176 | 8.3 ± 0.48 | 9.16 ± 0.48 | 9.46 ± 1.19 |
| Acute Inflammations[‡] | 184 | 0.0 ± 0.0 | 0.003 ± 0.006 | 0.0075 ± 0.012 |
| SPECTF Heart[‡] | 96 | 6.4 ± 1.24 | 6.67 ± 1.2 | 6.53 ± 1.29 |
| Glioma Grading Clinical & ...[‡] | 759 | 4.8 ± 0.75 | 6.43 ± 0.9 | 6.47 ± 0.55 |

## A.2 Impact of Data Dependent Factors

This section contains all the plots for impact of data dependent factors on Scikit-Learn's Pedregosa et al. (2011) simulated datasets. Lower area under the loss curve indicate better performance.

### A.2.1 Classification Datasets[4]

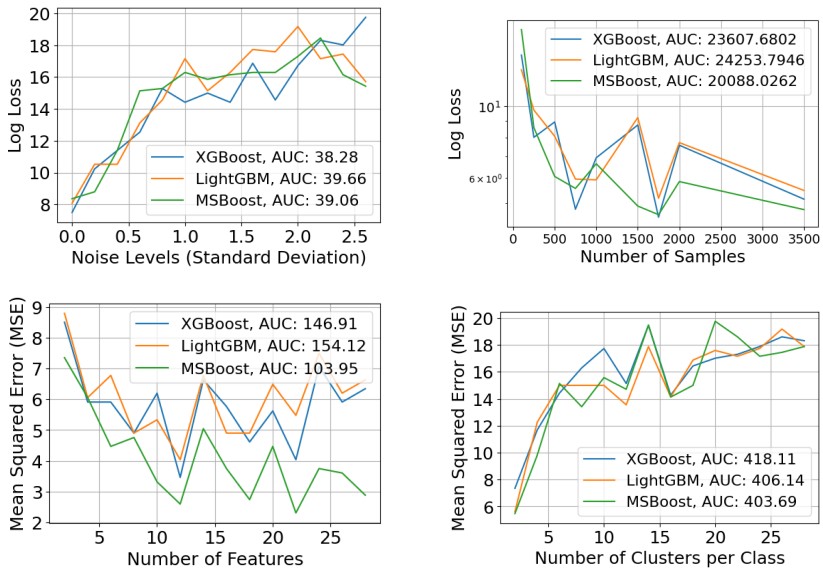

Figure 4: Make Classification Hastie et al. (2009b)

---

[4]Due to computational and hardware constrains the jupyter kernel crashed when the number of samples went more than around 5K, so it wasn't done on 10K samples like regression.

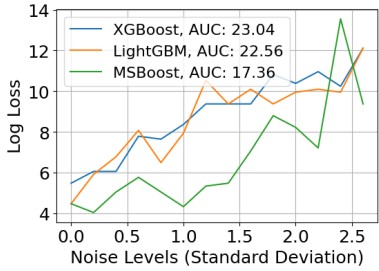 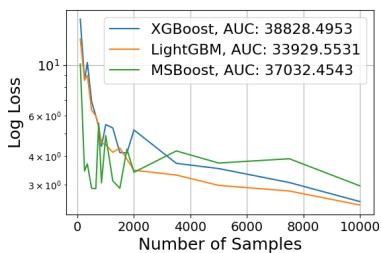

Figure 5: Hastie 10 Dataset Hastie et al. (2009b)

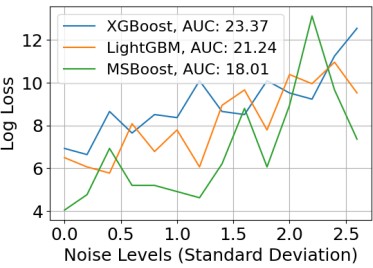 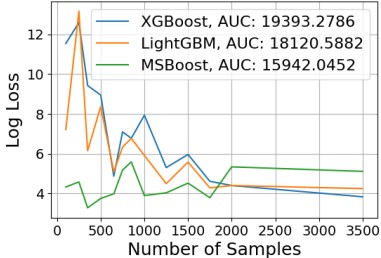

Figure 6: Gaussian Quantiles Hastie et al. (2009a)

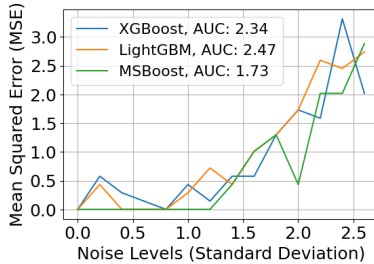 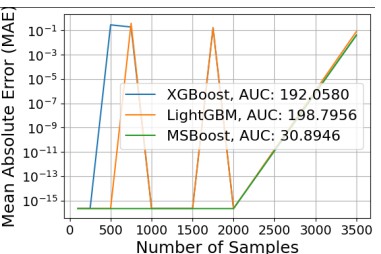

Figure 7: Make Blobs

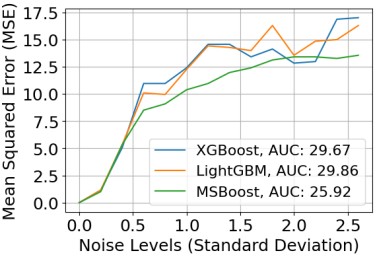 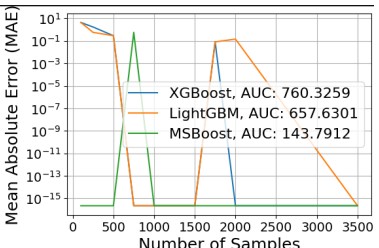

Figure 8: Make Moons

### A.2.2 Regression Datasets

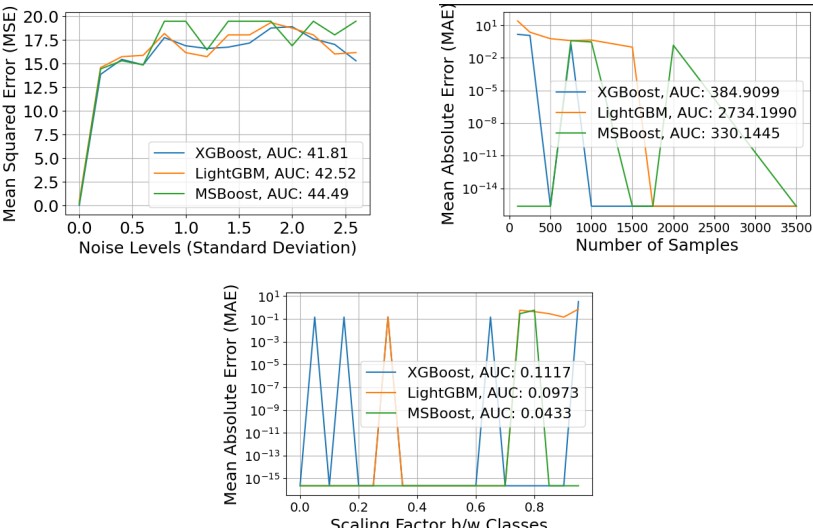

Figure 9: Make Circles

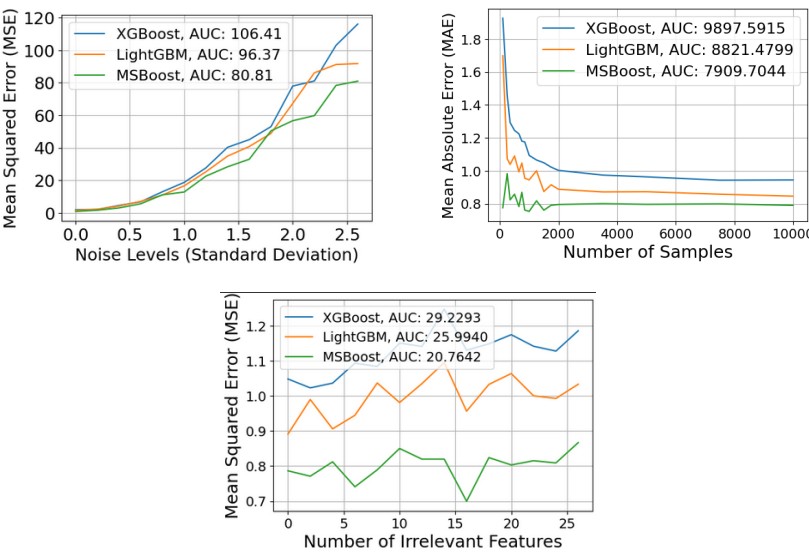

Figure 10: Sparse Uncorrelated Celeux et al. (2012)

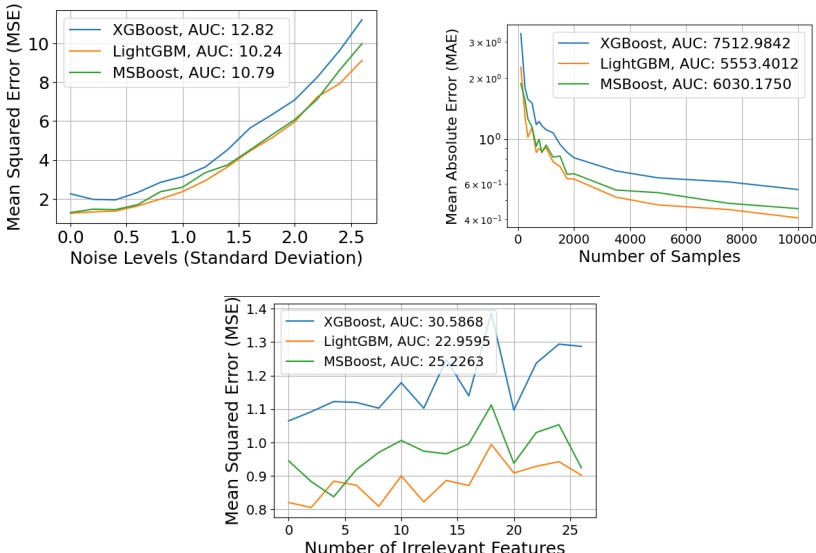

Figure 11: Friedman 1 Friedman (1991); Breiman (1996)

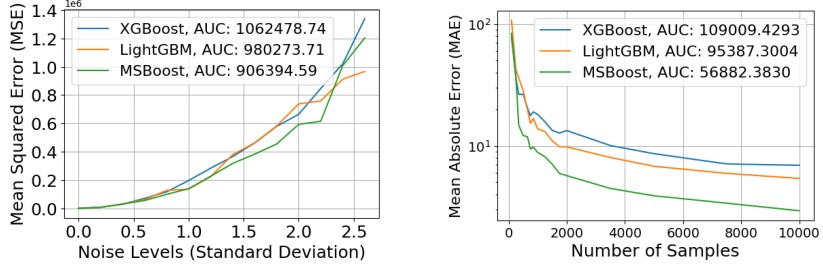

Figure 12: Friedman 2 Friedman (1991); Breiman (1996)

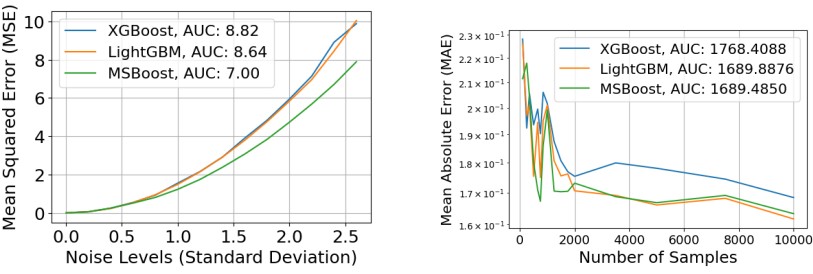

Figure 13: Friedman 3 Friedman (1991); Breiman (1996)

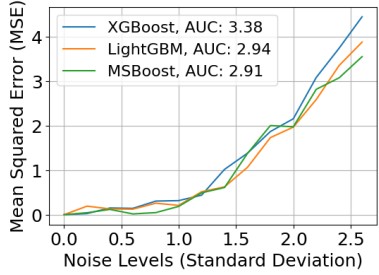 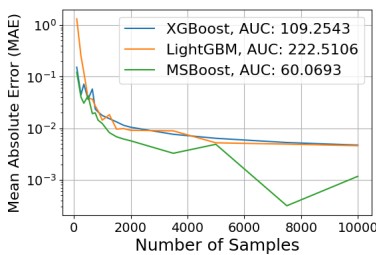

Figure 14: Swiss Roll Marsland (2011)

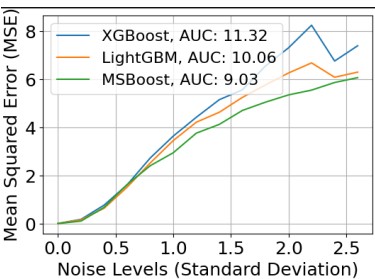 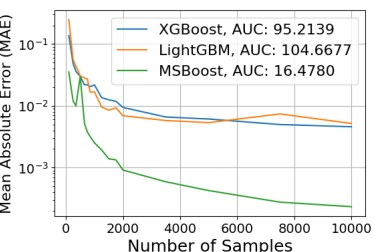

Figure 15: S Curve

## A.3 Impact of Model Selection Methods

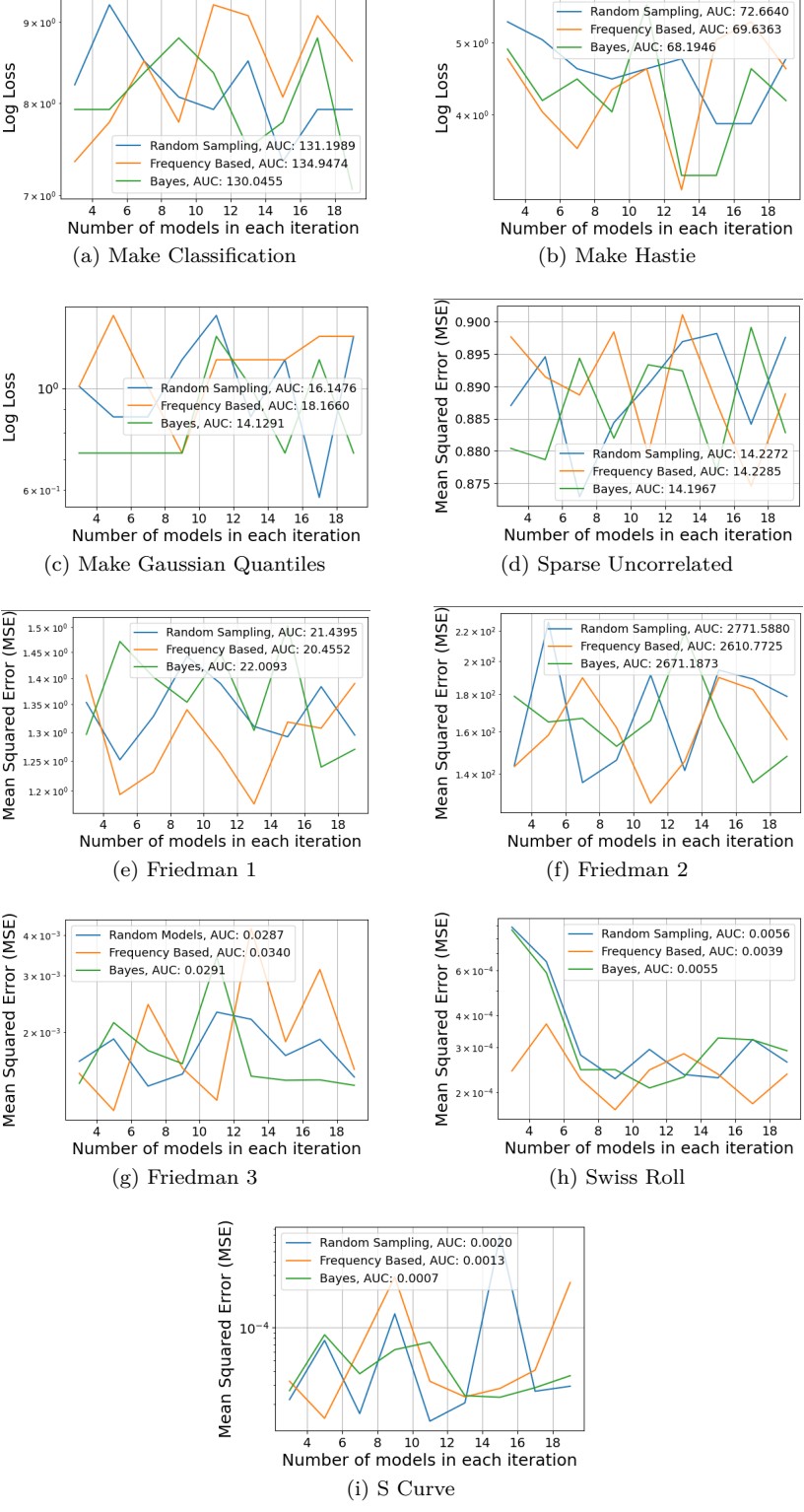

Figure 16: Impact of number of models for model selection methods

# B  Dataset Sources & Description[5]

## B.1  Benchmarking Datasets

This section contains descriptions for selected datasets used for benchmarking. Please refer to the original sources Vanschoren et al. (2014); Bischl et al. (2021); Pedregosa et al. (2011); Kelly et al. for descriptions for all the datasets used in Table 5 & 6.

- wave_energy: This data set consists of positions and absorbed power outputs of wave energy converters (WECs) in four real wave scenarios from the southern coast of Australia. The data is obtained from an optimization method (blackbox optimization) with the goal of finding the optimal buoys placement. Each instance represents wave energy returns for different placements of 16 buoys.

- Friedman 2: $y(X) = (X[:,0]^2 + (X[:,1] \times X[:,2] - \sqrt{\frac{1}{X[:,1]*X[:,3]))^2}} + \text{noise} \times N(0,1)$

- Sparse Uncorrelated: $X \sim N(0,1); y(X) = X[:,0] + 2 \times X[:,1] - 2 \times X[:,2] - 1.5 \times X[:,3]$

- kin8nm: A realistic simulation of the forward dynamics of an 8 link all-revolute robot arm. The task is to predict the distance of the end-effector from a target based on angular positions of the joints.

- sarcos: Dataset related to an inverse dynamics problem for a seven degrees-of-freedom SARCOS anthropo-morphic robot arm. Predict joint torques from joint positions, velocities, and accelerations.

- Moneyball: Dataset used in baseball analytics, focusing on statistics like on-base percentage (OBP) and slugging percentage (SLG) to predict player performance.

- yprop_4_1: Dataset used in the tabular data benchmark, transformed accordingly, for regression on categorical and numerical features.

- fps_benchmark: Dataset containing FPS measurements of video games executed on computers, characterized by CPU and GPU specifications and game settings.

- Zurich Transport: Zurich public transport delay data, cleaned and prepared for analysis.

- phoneme: Dataset to distinguish between nasal (class 0) and oral sounds (class 1) using harmonics and energy ratios.

- guillermo: The challenge introduces diverse, real-world datasets formatted uniformly for binary classification tasks, evaluated by AUC. Participants use preprocessed matrices and adhere to time-constrained evaluations on Codalab.

- MagicTelescope: Simulation data from a ground-based atmospheric Cherenkov gamma telescope, detecting high-energy gamma particles.

- heloc: Dataset used in the tabular data benchmark, transformed accordingly, for classification on numerical features.

- Bioresponse: Predict biological responses of molecules based on chemical properties and molecular descriptors.

- electricity: Dataset collected from the Australian New South Wales Electricity Market, containing 45,312 instances over a period from 7 May 1996 to 5 December 1998.

- Australian: Australian Credit Approval dataset, anonymized and converted to ARFF format, used in credit card application analysis.

- house_16H: Binarized version of the house dataset, converting numeric target features to a two-class nominal target feature based on mean values.

- pol: Dataset used in the tabular data benchmark for classification on numerical features, related to a telecommunication problem.

- california: The dataset includes data from all California block groups in the 1990 Census, averaging 1425.5 individuals per group in compact areas varying with population density. It features 20,640 observations across 9 variables, excluding groups with zero entries, with the dependent variable being ln(median house value).

---

[5]GPT-3.5 was used to summarize the data description from original sources.

## B.2 Social Impact Datasets

B.2 Social Impact Datasets

- AIDS Clinical Trials Group Study 175: The AIDS Clinical Trials Group Study 175 Dataset contains healthcare statistics and categorical information about patients who have been diagnosed with AIDS. This dataset was initially published in 1996. The prediction task is to predict whether or not each patient died within a certain window of time or not.

- Student Performance: The dataset analyzes student achievement in two Portuguese secondary schools, covering grades, demographics, and school-related factors. It includes separate datasets for Mathematics (mat) and Portuguese language (por), with a strong correlation between final grade (G3) and earlier grades (G2 and G1), essential for prediction and analysis according to Cortez and Silva (2008).

- Energy Efficiency: The dataset consists of 768 samples representing 12 different building shapes simulated in Ecotect. Variations include glazing area, distribution, orientation, and other parameters, generating 8 features per sample. The objective involves predicting two real-valued responses or, alternatively, using the rounded responses for multi-class classification.

- Diabetes: Contains 442 samples with 10 numeric features related to diabetes progression, including age, sex, BMI, blood pressure, and blood serum measurements. Target is a continuous measure of disease progression one year after baseline.

- Liver Disorders: The dataset contains records of male individuals with 5 blood test variables possibly related to liver disorders from alcohol consumption. The 7th field serves as a train/test selector, not as a dependent variable for liver disorder presence/absence; researchers should use the dichotomized 6th field (drinks) for classification.

- Heart Failure Clinical Records: This dataset contains the medical records of 299 patients who had heart failure, collected during their follow-up period, where each patient profile has 13 clinical features.

- Differentiated Thyroid Cancer Recurrence: This data set contains 13 clinicopathologic features aiming to predict recurrence of well differentiated thyroid cancer. The data set was collected in duration of 15 years and each patient was followed for at least 10 years.

- Rice (Cammeo and Osmancik): A study was conducted on Osmancik and Cammeo rice species, prominent in Turkey since 1997 and 2014 respectively. 3810 rice grain images were analyzed, deriving 7 morphological features per grain. Osmancik grains are noted for their wide, long, glassy, and dull appearance, while Cammeo grains exhibit similar characteristics with a focus on width and length.

- Blood Transfusion Service Center: This study utilized data from the Blood Transfusion Service Center in Hsin-Chu City, Taiwan, for a classification problem. The dataset comprises 748 donor records selected randomly, with features including R (Recency), F (Frequency), M (Monetary), T (Time since first donation), and a binary variable indicating blood donation in March 2007 (1 for donated, 0 for not donated). The objective was to develop a RFMTC marketing model using these variables.

- Acute Inflammations: The dataset was crafted by a medical expert to support an expert system for diagnosing two urinary system diseases: acute inflammation of the urinary bladder and acute nephritis. It utilizes Rough Sets Theory for rule detection, with each instance representing a potential patient.

- SPECTF Heart: Data on cardiac Single Proton Emission Computed Tomography (SPECT) images. Each patient classified into two categories: normal and abnormal. The dataset describes diagnosing of cardiac Single Proton Emission Computed Tomography (SPECT) images. Each of the patients is classified into two categories: normal and abnormal. The database of 267 SPECT image sets (patients) was processed to extract features that summarize the original SPECT images. As a result, 44 continuous feature pattern was created for each patient.

- Glioma Grading Clinical and Mutation Features: The dataset focuses on gliomas, primary brain tumors graded as LGG (Lower-Grade Glioma) or GBM (Glioblastoma Multiforme), based on histological/imaging criteria and molecular mutations. It includes the most frequently mutated 20 genes and 3 clinical features from TCGA-LGG and TCGA-GBM projects. The goal is to predict the glioma grade (LGG or GBM) using these features, aiming to identify the optimal subset for improved diagnostic accuracy and cost reduction in molecular testing for glioma patients.

