# OpenReview forum: "MSBoost: Using Model Selection with Multiple Base Estimators for Gradient Boosting"
_TMLR — Rejected by TMLR_

### Review · Reviewer_LvQR · 2024-11-02

**Summary Of Contributions:**

This paper presents MSBoost, a gradient boosting method that uses model selection to find the optimal base estimator at each boosting iteration. Traditional gradient boosting methods rely on a fixed model hypothesis class for the base learner, typically decision trees, to minimize residuals iteratively. MSBoost, however, selects from a pool of models at each iteration, choosing the one that best minimizes the validation error. Extensive empirical analysis suggests that MSBoost outperforms popular boosting algorithms (XGBoost, LightGBM) on several datasets. The authors also benchmark MSBoost on small noisy real-world datasets.

**Audience:**

Yes

**Broader Impact Concerns:**

Potential concerns have been addressed in the paper already

**Claims And Evidence:**

No

**Requested Changes:**

### Critical for securing recommendation

* Benchmark on a few (say atleast 5) larger datasets that have not been reduced to 1k samples
* Benchmark against other heterogenous boosting algorithms or alternatively, modify the scope of the claim to only include tree-based boosting algorithms like XGBoost and LightGBM
* For classification, report accuracies rather than log loss
* For most of the small noisy datasets, MSBoost's improvements are not statistically significant. Therefore, modifying the claim to reflect this would be appropriate.
* Information about what hypothesis classes are used during model selection
* All of the points noted under the review section `Clarity and correctness of presentation`

### Not critical but would strengthen the work
* Information about hyperparameters used for training all models.

**Strengths And Weaknesses:**

## Strengths
* The authors tackle the practical problem of learning predictors for tabular datasets with an additional focus on small noisy datasets that have real-world significance
* The empirical analysis carried out is extensive: the authors' method is compared with popular boosting algorithms on 50+ tabular datasets covering both regression and classification. They have also compared different model selection approaches. Ablative studies are included.

## Weaknesses

### Rigor in experiments

* Several of the datasets used in benchmarking are much larger than 1k samples in their original form. While I understand the reduction to 1k samples was done due to computational constraints, this reduces the robustness and reliability of these comparisons.
* All of the benchmarking for classification has been done by reporting log loss rather than accuracy, whereas the latter is of primary importance in classification. This is also important because the differences in log loss between methods can be very different than the differences in accuracy.
* In Table 3, for most datasets the standard deviations are too high to conclusively say MSBoost's improvements are statistically significant. It is also unclear why the absolute scores are not provided for baselines as done for Table 1 and 2.
* There is no comparison to other heterogenous boosting algorithms. [1] is one such example.
* For `wave_energy`, when LightGBM and XGBoost have MSEs on the order of `1e9`, it is surprising that MSBoost has an MSE close to `0`. I suspect this could be happening if MSBoost is using complex hypothesis classes during model selection. It is also unclear what the actual MSE is since it is reported as `0`.

### Lack of information in experiments

* There doesn't seem to be a mention of what model hypothesis classes the authors are sampling from during model selection. Given that model selection is a core aspect of this work I would have hoped to see that.
* How complex are the model hypothesis classes? (If these include neural networks for example, it would no longer be fair to compare MSBoost to only other tree-based gradient boosting methods. Morever, MSBoost would need a larger computational budget. Since the authors don't make a claim involving other model classes, this comparison is not a requirement, but this underpins why it is useful to provide information about the hypothesis class)
* There is no mention of hyperparameters used to train any of the models (both MSBoost and the baselines)

### Clarity and correctness of presentation

* Figure 1 is confusing because the models M1..Mn in a row are spread along the Iterations axis which is misleading.
* Figure 2 mentions AUC in the plot legend. If this is AUC-ROC, it is unclear why the AUC is greater than 1.
* Pg 2, the statement "which turns out to be the arithmetic mean of the target values" is incorrect as it depends on the loss function. While this is true for mean squared loss, this may not be true for other losses.
* Eq 2, it is unclear why there is an $r_i$ outside of the loss function
* Pg 2 last line: the final predictor is written as the summation of $F_i$, however it is unclear if this is intentional/correct because in general for boosting, the predictor is simply $F_n$
* Algorithm 1 Line 7: same issue as the previous point. Additionally, this expression is equated to 0. It is unclear what this means.
* Sec 2.1: $E$ is not defined
* Pg 2, the same variable $i$ is used to represent both sample index and boosting iteration
* Pg 6, in Limitations and Further Prospects (i), $n$ is not defined
* The word layer is used to denote a boosting iteration several times. This can be misleading as layer is more commonly used with layers in a neural network.

[1] SnapBoost: A Heterogeneous Boosting Machine; Parnell et al.

---

### Review · Reviewer_A8XJ · 2024-11-08

**Summary Of Contributions:**

The paper proposed a new boosting algorithm which uses model selection at each update of the model, if I understand the paper correctly.

**Audience:**

No

**Broader Impact Concerns:**

No much concern.

**Claims And Evidence:**

No

**Requested Changes:**

The paper should use mathematically rigorous notations. All notations should be defined rigorously before their use.

The proposed algorithm should be compared with the standard boosting algorithm explicitly to highlight the main contribution of the proposed algorithm.

Theoretical evidences should be provided. Usually, the expansion of the set of base learners would not be helpful since
the boosting model locates in the linear span space of base learners, and the linear span space of base learners could be
quite large even when the set of base learners is quite small.

**Strengths And Weaknesses:**

1. The paper is not well written. In fact, I could not understand what are differences of the proposed MS boost and the standard Gradient boosting. For me, Algorithm 1 is exactly the same as the standard gradient boosting.

2. I could not understand the three model selection methods proposed in Section 2.2. In the naive method, it is written that 'to train all the available base estimators in parallel'. The boosting algorithm is a sequential algorithm. I cannot imagine how all base estimators are updated simultaneously in the boosting algorithm.

3. The third model selection method - 'Frequency and Probability Based sampling' seems to be too heuristic. First of all, I could not see why such biased sampling improves the performance. In addition, it seems to use the posterior probabilities which are not defined properly. To use the term 'posterior probability', there must be model and prior and so the posterior probability can be defined by use of the Bayes theorem.  Also, there are many notations not defined properly. Examples are E, T, s_i and so on.

---

### Review · Reviewer_JTym · 2024-11-27

**Summary Of Contributions:**

The manuscript „MSBoost: Using Model Selection with Multiple Base Estimators for Gradient Boosting” presents a novel approach for gradient boosting. The idea is to not just a single model for the boosting (e.g., a decision tree), but rather try out different models (e.g., SVM, decision tree, …) and during each iteration select the model that reduces the residual error the most.

**Audience:**

Yes

**Broader Impact Concerns:**

I suggest to possibly drop this statement, as it is rather generic for the development of any new ML model.

**Claims And Evidence:**

No

**Requested Changes:**

For the above reason, I suggest the following improvements. I deem all of them to be critical:
- Report and evaluate the runtime for all experiments
- Evaluate what happens when other boosting variants have the same compute budget as MSBoost
- Provide details regarding the used base models

**Strengths And Weaknesses:**

While I like the idea, I think further experimentation is required to show the merits of this approach, as the execution time is currently not considered. Notably, boosting typically relies on rather weak learners to deal with the execution time issue and then stack many of these weak learners. MSBoost breaks this, by allowing strong learners at each layer, while keeping the number of layers fixed. Details regarding these problems can be found below.

1) The authors state that the time is O(Iterations x base model with highest complexity). While this may be technically true, I think this is misleading, as this seems like the effort is “just” the time complexity of the worst model for each iteration. However, the constants at play here are possibly huge. E.g., if the difference between the base model with the highest complexity and others is not large, the runtime would be more or less linear in the number of base models as well.
That this is an important consideration is shown by the approach itself: if it really would be like the reported complexity and, mostly, only depend on a single base model per iteration, the random sampling or frequency approach would have a similar run-time than the naïve approach (assuming that the base-model with highest complexity is regularly selected). However, these are specifically introduced to deal with runtime issues.
In my opinion, this needs to be discussed more carefully.

2) Directly related to the above is that the suggested approach is very expensive in terms of computational effort. Nevertheless, no runtime estimates are reported. Consequently, it is unclear how much more expensive the naïve approach is in comparison to the other variants for model selection (Sec. 2.2), nor do we know how much more expensive this is in comparison to the baselines, i.e., LightGBM and XGBoost.
Reporting of performance is crucial to understand the feasibility of the approach, as this is already reported as a notable limitation and only data sizes of up to 1k are used.

3) Another runtime related question is whether MSBoost would be better, if the same compute budget is available. Based on the experiment code, it seems like all models are trained with 100 estimators. My guess is that a lot larger models are possible for LightGMB and XGBoost in the time it takes to train MSBoost.
The experiments need to be extended with a comparison that is fair in terms of run-time used such that LightGMB and XGBoost are on an even playing field.
(Note that this is considering only training time. Given equal performance (or even worse for with MSBoost), when this is the same for all models, one could still possibly argue in favor of MSBoost, in case prediction times are faster.)

4) The candidate models used by MSBoost are not mentioned in the paper. In the reported data, there are actually three sets of such models (at least for classification, did not check the others): complex, simple, and default. It is unclear how any of these were selected and used during the development and experimentation. This needs to be made transparent. Moreover, it seems like reporting performance (and runtime) differences for different base models would be a valuable ablation to support the model choice. Finally, one of the base models in the complex set is actually a GradientBoostingRegressor. Related to point three, this seems a bit unfair: the compute budget of a complete boosting model for free in a single boosting iteration. This shows why I believe more data regarding this is crucial.

5) There is also no reporting *which* base models are currently selected by MSBoost. This should be reported, ideally also including at which layer of the boosting. This would yield insights regarding the importance of different base models.

6) There are many typographical issues. Most notably missing braces for references, but also some uncommon or incomplete sentence structures (e.g., second sentence of the paper that is only “Such as” with a list of examples). These should be fixed in another round of proof-reading.

---

### Decision · Action_Editor_itxu · 2024-12-23

**Recommendation:** Reject

**Comment:**

Based on the comments from the reviewers and having no discussion / revision from the authors side, the paper seems to be in it earlier draft and needs to be reworked in different aspects to answer critical questions raised by reviewers before resubmission:
- clarity of presentation, including typos, rigor in mathematical notation, clear explanation of the proposed method and comparison with prior work (all reviewers).
- clarity and deep discussion of empirical setup, including class of base learning, hyperparameters (all reviewers).
- correctness of empirical validation is under question (pointed out by different reviewers):
  - issue metric: reporting only logloss and not the accuracy (or ROC AUC for binary classification e.g.) cannot validate the method.
  - improvements are not statistically significant, which raises the question for the need to bring the new method to the table.
  - validation of the method on large data is needed.
  - proper baselines, comparison with them and highlighting the difference with them should be provided.
- theoretical validation
  - all notation should be clear to be able to assess the method.
  - if possible (if not then discuss maybe why) provide theoretical evidence why the proposed method should work better than vanilla gradient boosting (see points from the reviewer A8XJ on that).
- runtime issue
  - as pointed out by the reviewer JTym the critical issue with the proposed method is speed and slow down compared to vanilla boosting (which is optimized specifically for tree-based base learners).
  - thus runtime should be provided for practical applications.
  - comparison with the fixed compute budget between different variants of boosting and the proposed method should be provided to justify higher compute budget (if any).

I strongly recommend authors to revisit and rework the paper by taking into account the above concerns before resubmitting it.

**Audience:**

Gradient boosting methods are widely used for tabular data nowadays. The paper targets the question how to improve the gradient boosting by considering another class of base learners, which will be interesting to the TMLR audience from various perspectives, e.g. application to tabular data; understanding dependence on the base learners class.

**Claims And Evidence:**

The paper proposes a new gradient boosting variant. While the idea is interesting, the major issues with the paper are absence of theoretical justification, very weak empirical justification to show that the proposed method is better (statistically, by important metrics used in classification, by proper datasets, by proper baselines). Also notation, clarity of the presentation and comparison with prior work need the work before publication.

In summary, both theoretical and empirical results do not have enough support to justify the need, validity and improvements of the proposed method -- details see below in the "Comments" section.

**Resubmission Of Major Revision:**

The authors may consider submitting a major revision at a later time.